# Applicability of Oculomotor Tests for Predicting Central Vestibular Disorder Using Principal Component Analysis

**DOI:** 10.3390/jpm12020203

**Published:** 2022-02-02

**Authors:** Ching-Nung Wu, Sheng-Dean Luo, Shu-Fang Chen, Chi-Wei Huang, Pi-Ling Chiang, Chung-Feng Hwang, Chao-Hui Yang, Chun-Hsien Ho, Wei-De Cheng, Chung-Ying Lin, Yi-Lu Li

**Affiliations:** 1Department of Otolaryngology, Kaohsiung Chang Gung Memorial Hospital and Chang Gung University College of Medicine, Kaohsiung 83301, Taiwan; taytay@cgmh.org.tw (C.-N.W.); rsd0323@cgmh.org.tw (S.-D.L.); cfhwang@cgmh.org.tw (C.-F.H.); chouwhay@gmail.com (C.-H.Y.); jayjaybrot@cgmh.org.tw (C.-H.H.); Wei.de.cheng@gmail.com (W.-D.C.); 2Department of Public Health, College of Medicine, National Cheng Kung University, Tainan 701401, Taiwan; 3Graduate Institute of Clinical Medical Sciences, College of Medicine, Chang Gung University, Taoyuan 33302, Taiwan; 4Department of Neurology, Kaohsiung Chang Gung Memorial Hospital and Chang Gung University College of Medicine, Kaohsiung 83301, Taiwan; fangoe@cgmh.org.tw (S.-F.C.); justin1124@cgmh.org.tw (C.-W.H.); 5Department of Diagnostic Radiology, Kaohsiung Chang Gung Memorial Hospital and Chang Gung University College of Medicine, Kaohsiung 83301, Taiwan; lovage@cgmh.org.tw; 6Institute of Allied Health Sciences, College of Medicine, National Cheng Kung University, Tainan 701401, Taiwan; 7Department of Occupational Health, College of Medicine, National Cheng Kung University, Tainan 701401, Taiwan; 8Biostatistics Consulting Center, National Cheng Kung University Hospital, College of Medicine, National Cheng Kung University, Tainan 701401, Taiwan; 9Department of Otolaryngology, National Cheng Kung University Hospital, College of Medicine, National Cheng Kung University, Tainan 701401, Taiwan; 10Institute of Clinical Medicine, College of Medicine, National Cheng Kung University, Tainan 70101, Taiwan

**Keywords:** central vestibular disorder, saccade, pursuit, gaze evoked nystagmus, principal component analysis

## Abstract

The videonystagmography oculomotor test battery is considered useful method for diagnosing vertigo. However, its role in diagnosing central vestibular disorder has not been clarified due to variations in interpretation. Patients (*n* = 103) with vertigo or dizziness symptoms undergoing the oculomotor tests and brain MRI within 1 month were analyzed. Two otology specialists retrospectively interpreted the oculomotor tests, and three neurology and neuroradiology specialists determined whether central lesions were present on brain MRI. Multivariable logistic regression analysis was performed to determine the factors contributing to discordant interpretation between oculomotor tests and brain MRI. Oculomotor tests predicting central lesions were assessed using principal component analysis. The intra- and inter-rater reliability in oculomotor test interpretation was moderate to good. Age > 60 years and multiple comorbidities were significant predictors of a discordant interpretation between MRI and oculomotor tests. Positive neurological symptoms and a higher oculomotor index (according to saccade (vertical axis), smooth pursuit (horizontal axis), and gaze-evoked nystagmus (horizontal/vertical axes) tests) significantly predicted central vestibular disorder in vertigo patients. Caution is required when interpreting the results of the oculomotor test battery for diagnosis of central lesions in older patients, as well as in those with multiple comorbidities.

## 1. Introduction

Vertigo is a common but multifactorial disorder with a lifetime incidence of 20–30% [1]. It can be classified as peripheral or central in origin. Central vestibular disorders, such as infarction/transient ischemic attacks in the vascular territories of the posterior fossa and brain tumors involving retrocochlear pathways, are responsible for 25% of cases of acute vertigo [2,3]. Central vertigo may have diverse clinical manifestations depending on the underlying diseases, and about 20% of patients present with isolated vertigo rather than focal neurological deficits, making the diagnosis difficult [4,5,6]. As part of daily clinical practice for otolaryngologists, neurologists, and emergency physicians, there are still no reliable, prompt, and efficient screening methods for this condition, thus necessitating comprehensive assessment.

A tentative diagnosis can be made through detailed history-taking and physical examinations, including neurotological findings. Nystagmus with central features detected via bedside oculomotor examination indicates a central origin [7,8]. As the gold standard diagnostic imaging modalities, computed tomography (CT) and magnetic resonance imaging (MRI) can reveal intracranial lesions in cases of suspected central vertigo [9]. As most abnormalities, such as cerebral atrophy and white matter lesions, are equally common in people with and without dizziness, the routine arrangement of MRI is unhelpful and unnecessary for determining the specific cause of dizziness or vertigo [10]. Furthermore, MRI studies yield only 12.2% positive findings of significant abnormalities in vertigo patients with considerable costs in most health systems worldwide [11]. Regarding a relatively low cost-effectiveness of MRI scans, vestibular function tests may play an important role as screening tests supporting imaging studies, thus facilitating clinical decision making.

Videonystagmography (VNG) is considered a useful method for diagnosing vertigo of peripheral origin [12]. VNG has been widely adopted over the past few decades, as it is less time-consuming and shows higher diagnostic accuracy than electronystagmography (ENG). VNG involves a series of subtests of the responses of the vestibular end organs, central vestibuloocular pathways, and oculomotor processes, on the basis of directly recording eye movements. However, some tests in the VNG battery, such as positional and caloric irrigation tests, can be difficult for patients to tolerate because of the powerful sense of vertigo induced by the stimulus [13]. Oculomotor tests cause less discomfort during testing; the only problem regarding their diagnosis of central vestibular disorder is the subjective interpretations of the results [14,15]. Therefore, understanding the consistency in interpretations of oculomotor tests is important for clinicians in deciding whether they should be used.

This study was performed to examine the utility of oculomotor tests for diagnosing central vestibular disorder on the basis of the consensus of multiple interpreters. Further, we aimed to identify correlations of clinical manifestations and oculomotor test results with abnormal lesions on brain MRI.

## 2. Materials and Methods

### 2.1. Study Design and Population

A retrospective study was carried out in patients with vertigo or dizziness symptoms undergoing oculomotor tests and referred for brain MRI for further assessment between September 2013 and August 2020 in a tertiary hospital. Patients with more than 1 month between oculomotor tests and brain MRI were excluded to prevent any bias associated with disease progression. Demographic and clinical characteristics were extracted from medical records, including age; sex; body mass index (BMI); history of underlying disease; and the results of neurotological, audiometry, oculomotor, and brain MRI tests.

This study was divided into two parts (Figure 1). First, the utility of oculomotor tests was examined. Specifically, we checked for consistency in the interpretation of oculomotor tests conducted by two otology specialists. We also determined factors affecting the utility of oculomotor tests for predicting central lesions. Second, a central vestibular disorder study was conducted. The characteristics of the patients and oculomotor tests were analyzed to determine whether there were factors associated with findings of central lesions on brain MRI. This study was approved by the institutional review board of Chang Gung Medical Foundation (approval no. 201901052B0 and 202001115B0).

### 2.2. Oculomotor Tests

A VNG test battery was performed using VNG Ulmer system (Synapsys, Mountain View, CA, USA); this includes a number of tests, including of spontaneous nystagmus with or without fixation, gaze-evoked nystagmus, positional and positioning nystagmus, saccades, smooth pursuit, and optokinetic nystagmus, as well as caloric testing. In this study, only the oculomotor tests, including eight dimensions (gaze-evoked nystagmus (horizontal/vertical axes), saccades (horizontal/vertical axes), smooth pursuit (horizontal/vertical axes), and optokinetic nystagmus/afternystagmus), were measured for an individual subject.

To minimize subjectivity, two otology specialists interpreted all of the test results. The results of the oculomotor study were first classified as either indicative of central vestibular disorder or nonspecific findings. Oculomotor signals within individual dimensions were labeled abnormal if they indicated central vestibular disorder and labeled normal if showing nonspecific findings. Both specialists performed the oculomotor studies again after 1 month so that we could assess intra-rater reliability. Central vestibular disorder was considered present on the basis of the results of the following tests, which the consensus of detailed criteria had been reached: [16]
Gaze-evoked nystagmus (horizontal or vertical axis): The patient was made to look front, left, right, up, and down at angles of 15° for 20 s at one side, and the nystagmus was recorded. The whole interpretation range for one dimension was 60 s. Pure torsional or vertical nystagmus, direction-changing nystagmus, and gaze-evoked nystagmus opposite to Alexander’s law were considered as central vestibular findings.Saccades (horizontal or vertical axis): A sequence of spots displaced horizontally or vertically at roughly 4 s intervals were shown, and the patient was asked to follow the stimulus only with their eyes, keeping their head stable. The whole interpretation range for one dimension was 30 s, including 8–9 repetitive signals. Latency refers to the delay between the onset of target movement and initiation of eye movement; latency consistently >260 ms is considered abnormal. Precision refers to the amplitude of the eye movement relative to the target; hypometria of 10–20% is considered normal. Asymmetrical saccades indicate abnormality.Smooth pursuit (horizontal or vertical axis): Patients watched a bright spot that moved smoothly across a screen in the horizontal or vertical plane, at a frequency of 0.25 Hz. The whole interpretation range for one dimension was 30 s, including 7–8 repetitive signals. Pursuit may be saccadic/jerky if there is a lesion in the cerebellum, brainstem, or parietal lobe. “Symmetrically impaired” pursuit can be a nonspecific central finding, while “asymmetrically impaired” pursuit is suggestive of a unilateral hemispheric or asymmetrical posterior fossa lesion.Optokinetic nystagmus: The patient was asked to track a repetitive moving target across the screen at a frequency of 4 Hz. These were moved first toward the right and then toward the left. The whole interpretation range for optokinetic nystagmus was 15 s, and another 15 s for after nystagmus reading as well. Cerebral and cerebellar lesions lead to ipsilateral preponderance, and brainstem lesions to contralateral preponderance.

### 2.3. Brain MRI

All patients had undergone multiplanar MRI of the brain with multiple pulse sequences. All MRI scans were performed on the 3-Tesla GE Signa whole-body MRI system (General Electric Healthcare, Milwaukee, WI, USA), equipped with an eight-channel head coil. The MRI sequences included T1 weighted, T2 weighted, fluid attenuated inversion recovery (FLAIR), T2*, apparent diffusion coefficient (ADC), diffusion weighted imaging (DWI), and magnetic resonance angiography (MRA). Two neurology specialists interpreted the brain MRI findings in this study; this modality is the gold standard to determine the presence or absence of central lesions. A third neuroradiology specialist made the final decision on the MRI results if there were inconsistencies in interpretation between the two neurologists. The different diagnoses depended on the combination of the serial MRI contrasts presentation. For example, the cerebral infarction and hemorrhage were according to patient history (onset: acute, subacute, or chronic and neurological examinations), then to find out the corresponding DWI, ADC, T1 weighted, T2 weighted, FLAIR, gradient echo T2 weighted, and MRA lesions contributing the vessels’ territory. The disease identification was made by visual estimation toward the combination of serial MRI contrasts. Central lesion characteristics associated with vertigo were determined. Cerebrovascular events (ischemia/hemorrhage), tumors, and inflammation were recorded, along with lesion sites (including the cortex, subcortical regions, brainstem, and cerebellum).

### 2.4. Data Analysis

The demographic and clinical characteristics of the patients, as well as the characteristics of the central lesions, were summarized using descriptive statistics. We determined the sensitivity, specificity, positive predictive value (PPV), and negative predictive value (NPV), with corresponding 95% confidence intervals (CIs), of the oculomotor tests for central vestibular disorder diagnosis. To assess intra- and inter-rater reliability between the two raters, we calculated the kappa statistic. To determine the factors associated with consistency between the oculomotor tests and brain MRI, we analyzed categorical data (e.g., sex, hearing loss status, neurological symptoms, central lesions, and comorbidities) using the two-sided Fisher’s exact test or two-sided Pearson’s chi-squared test. Non-parametric continuous data (e.g., age and BMI) were analyzed using the Mann–Whitney *U* test. A multivariate logistic regression model was used to identify significant predictors of inconsistency between oculomotor and brain MRI tests.

Univariate logistic regression was used to identify dimensions of oculomotor tests significantly predicting central lesions. Dimensions classified as abnormal by either rater were coded them as “1”, while those not classified as abnormal were coded as “0”. Due to the moderate correlations between interpretation (normal/abnormal) of different dimensions, we performed principal component analysis (PCA) of the significantly predictive dimensions to construct an oculomotor index using the coding. The weight of each test was determined according to factor loading. The oculomotor index was computed as the sum of the product of the factor loadings and values of the individual tests. Finally, to further adjust out the effects of confounding factors and interactive relationship between variables, we assessed the ability of clinical characteristics and the oculomotor index to predict central lesions on MRI using a multivariate logistic regression model. All statistical analyses were performed using SAS software (version 9.4; SAS Institute Inc., Cary, NC, USA). In all analyses, *p* < 0.05 was taken to indicate statistical significance.

## 3. Results

The demographic and clinical characteristics of the 103 patients enrolled in this study with vertigo or dizziness symptoms, and undergoing oculomotor tests and brain MRI, are summarized in Table 1. These characteristics included age, sex, BMI, comorbidities, and symptoms associated with the most recent episode, such as asymmetrical hearing loss and other neurological symptoms. Comorbidities were considered to reflect the severity of underlying disease and were thus included in the analysis.

Twenty-four patients were diagnosed with central vestibular disorders on the basis of the results of brain MRI. The brain MRI lesions in the combination of the serial contrasts defined as abnormal corresponding to the nystagmus has been associated with the structure lesion involving cortical eyefields, subcortical, cerebellar, brain stem, and skull base [17,18]. Cerebrovascular accident (CVA; either ischemic or hemorrhagic) accounted for the largest proportion of central vestibular disorder cases (54.2%), followed by brain tumor (37.5%) and inflammation (8.3%). The most frequent sites of central lesions in this group were the cortical and subcortical areas (38.5%), brainstem (pontine and medulla) (38.5%), cerebellar area (19.2%), and skull base. Two sites occupied simultaneously by one lesion was noted in two patients. The only significant differences between the two groups were the sex ratio and associated neurological symptoms.

Two otology specialists interpreted the oculomotor test results (gaze-evoked nystagmus, saccades, smooth pursuit, and optokinetic nystagmus) of all 103 patients (Table 2). The intra-rater reliability, assessed after all of the oculomotor studies had been repeated 1 month later, was moderate to good (κ = 0.571 and 0.669 for the two raters). The sensitivity of the oculomotor tests for predicting central lesions ranged from 54.2% to 66.7%, and the specificity ranged from 43% to 67.1%. The inter-rater reliability was moderate (κ = 0.480), indicating some subjectivity in the interpretation of the oculomotor test results.

Only oculomotor test data on which the two raters agreed were included in further analysis. In total, test results among 75 patients were analyzed; 44 of these showed consistency with the results of brain MRI (Table 3). Significant differences between the groups with consistent versus inconsistent oculomotor tests and MRI findings were only observed for age and comorbidities. Univariate and multivariate logistic regression analyses were performed to determine predictors of inconsistency (Table 4). Age ≥ 60 years and having more than one comorbidity were significantly associated with inconsistency in the interpretations of oculomotor tests and brain MRI in univariate analysis. In multivariate analysis, age ≥ 60 years was again significantly associated with inconsistent interpretation (odds ratio (OR) = 3.09, 95% CI = 1.04–9.14). Multiple comorbidities (≥3) showed a trend toward a significant association with inconsistent interpretation (OR = 8.31, 95% CI = 0.96–71.7).

Univariate analysis also showed that abnormal findings for the vertical axis in the saccade test, for the horizontal axis in the smooth pursuit test, and for both the horizontal and vertical axes in the gaze-evoked nystagmus test, significantly predicted the presence of central lesions on brain MRI (Table 5). Because moderate correlations were observed between the four dimensions of tests (r = 0.318 to 0.575), PCA was performed to extract principal components from the four predictive tests for further multivariate analysis. We examined the first principal component (PC) 1 (Figure 2A). The eigenvalue was 2.34, and this component explained 58.6% of the total variance. The biplot demonstrates that all of four oculomotor tests were positively correlated and contributed primarily to PC1 (Figure 2B). This suggests that PC1, termed the oculomotor index, can serve as a weighted summary score of all oculomotor tests (the formula is given below). Therefore, a higher PC1 score (again, oculomotor index) is associated with more abnormalities on the oculomotor tests. Moreover, central lesions evident on brain MRI were well predicted by the oculomotor index (OR = 1.72, 95% CI = 1.24–2.38).
Oculomotor index = 0.505 Saccade_V + 0.519 Pursuit_H + 0.515 Gaze_H + 0.459 Gaze_V(1)

The results regarding clinical characteristics and the oculomotor index as predictors of central lesions in the 103 patients are shown in Table 6. The oculomotor index was divided at the 50% level, corresponding to fewer (below 50%) and more (above 50%) abnormalities evident on oculomotor testing. Although univariate analysis showed that male sex, positive neurological symptoms, and the top 50% of the oculomotor index scores were associated with central vestibular disorder, after adjusting for age and sex, only positive neurological symptoms (OR = 13.45, 95% CI = 4.00–45.12) and the top half of the oculomotor index scores (OR = 4.59, 95% CI = 1.28–16.44) significantly predicted central vestibular disorder on multivariate logistic regression analysis. In brief, vertigo patients with positive neurological symptoms and more specific abnormal oculomotor findings tended to exhibit central vestibular disease.

## 4. Discussion

To our knowledge, this was the first study to investigate the utility of oculomotor tests for diagnosing central vestibular disorder according to a consensus between multiple interpreters. Older patients (≥60 years) and those with more than one comorbidity were more likely to have oculomotor test results discordant with brain MRI. Furthermore, vertigo patients with positive neurological symptoms and greater numbers of abnormal oculomotor findings should be examined in detail for central vestibular disorder.

The present study was performed to determine the value of oculomotor tests for predicting central lesions on MRI in patients seen at a tertiary referral center. This was the first study to estimate the reliability and consistency of the interpretations of oculomotor tests between two otology specialists. This was important. The inter-rater reliability was (unsurprisingly) only moderate (κ = 0.480) (Table 2). The interpretations of oculomotor tests depend on rater experience; caution is required when using only a single rater. 

Furthermore, this study showed that oculomotor tests, regardless of rater, had only moderate sensitivity (50–70%) and specificity (40–70%) for predicting central lesions on MRI, in line with previous findings [14,15,19]. Interestingly, both raters showed low PPV but high NPV, suggesting the necessity of imaging studies if clinicians cannot confidently exclude central vestibular disorder. PPV and NPV are important indicators of the utility of a screening test for the general population [20]. The high NPV in this study indicates that negative results can be helpful to exclude central vestibular disease, while the low PPV suggests an increased likelihood of false-positive findings when oculomotor tests show positive results, especially given the low incidence of central vestibular disease in the study population. According to the above considerations, it seems that oculomotor tests are still of value for evaluating central vestibular disorders, particularly in terms of ruling out certain diagnoses. 

To clarify the factors underlying discordance in the interpretation of oculomotor tests and brain MRI results, we only included test data that were interpreted similarly by the two raters (*n* = 75) in the analysis. The results showed that age ≥ 60 years and comorbidities significantly predicted discordant interpretations (Table 4). In the older population, self-reported symptoms are more likely to include nonspecific dizziness and instability, and less likely to include rotary vertigo, relative to younger patients [21]. Previous reports also discussed factors associated with nonspecific dizziness and imbalance with advanced age, including sensory deficits, impaired visual acuity, sedative or antihypertensive drugs, musculoskeletal problems, deficits in postural control, and anxiety, which may be exacerbated by the presence of multiple comorbidities [22,23]. Good performance in an oculomotor test is denoted by the ability to trace specific targets. According to the above, aging and multiple comorbidities may be related to poor performance in oculomotor tests, and an increased likelihood of an incorrect diagnosis; this was also the case when the results for each rater were considered separately (Table 7). The validity of oculomotor tests, especially the specificity, decreased significantly in patients aged ≥ 60 years, as well as in those with multiple comorbidities. Therefore, caution is required when using the results of oculomotor tests to diagnose central vestibular disorder in older patients and those with multiple comorbidities.

Another valuable finding of the present study was that neurological symptoms and an abnormal oculomotor index strongly suggest central lesions in. A total of 16 of 24 patients with central vertigo presented with neurological symptoms (66.7%), and only one had asymmetric hearing loss (8.3%). Recognition of neurological symptoms, including dysphagia, diplopia, slurred speech, facial paralysis, sensory deficit, and limb weakness, is required before further management. Here, we also identified positive findings in vertical saccade, horizontal pursuit, and gaze-evoked nystagmus tests as predictors of central vestibular disorder, according to PCA (Table 5). This was generally consistent with previous studies indicating that oculomotor abnormalities in saccade, pursuit, gaze, and optokinetic tests are associated central vestibular disorders [12,24]. Different locations of central lesions lead to different clinical central vestibular findings in oculomotor tests, which may be responsible for the discrepancies in predictive value among studies. Impairments in pursuit and optokinetic tests may result from brainstem, cerebellar, and cerebral lesions, while abnormal saccades are often attributable to lesions in the paramedian pontine reticular formation, brainstem, cerebellum, and basal ganglia [12,14,24,25]. The central lesions identified by MRI in this study involved the cortical and subcortical areas, brainstem (pontine and medulla), and cerebellum, and were also correlated with abnormalities in oculomotor tests. The inter-rater reliability was only moderate, but our results nevertheless indicate that, after PCA, saccade (vertical axis), smooth pursuit (horizontal axis), and gaze-evoked (horizontal/vertical axes) nystagmus tests predict central vestibular disorders in vertigo patients. Multivariate logistic regression, which attempts to control for covariance, may discard certain domains that exhibit elements of collinearity. PCA is an alternative approach that deals with multicollinearity by expressing covariates as independent PCs [26]. We used PCA to identify the predictive utilities of specific oculomotor tests. An additive effect was confirmed via multivariate logistic regression; higher numbers of abnormalities apparent on specific dimensions of oculomotor testing increased the possibility of central vestibular disease. In fact, to calculate a correct coefficient of PCA in our study was impractical in the real world. The practical application of the formula was that more abnormal labels of the four dimensions in oculomotor tests indicated higher possibility of central lesions found in brain MRI.

This study had several limitations. First, we did not evaluate in detail how the oculomotor signals in the individual tests, such as saccadic pursuit, predicted central vestibular disorder. Therefore, further analysis of these signals was not performed. Second, oculomotor tests alone may be insufficient for diagnosis of central lesions, and other components of the VNG battery may also be necessary. However, this study was performed to determine specifically the diagnostic utility of the oculomotor tests. As mentioned above, the oculomotor tests were useful for evaluating central vestibular disorders, principally because they ruled out certain diagnoses. The major strength of this study was that the conclusions were derived from a consensus between multiple interpreters, which decreased the subjectivity of the interpretations of the oculomotor tests and MRI findings.

## 5. Conclusions

Oculomotor tests are useful to diagnose central lesions in vertigo patients. However, caution is required when diagnosing older adults and patients with multiple comorbidities. Vertigo patients with positive neurological symptoms and more abnormal oculomotor findings on the vertical axis in the saccade test, on the horizontal axis in the smooth pursuit test, and on both the horizontal and vertical axes in the gaze-evoked nystagmus test require detailed assessment for central vestibular disorders.

## Figures and Tables

**Figure 1 jpm-12-00203-f001:**
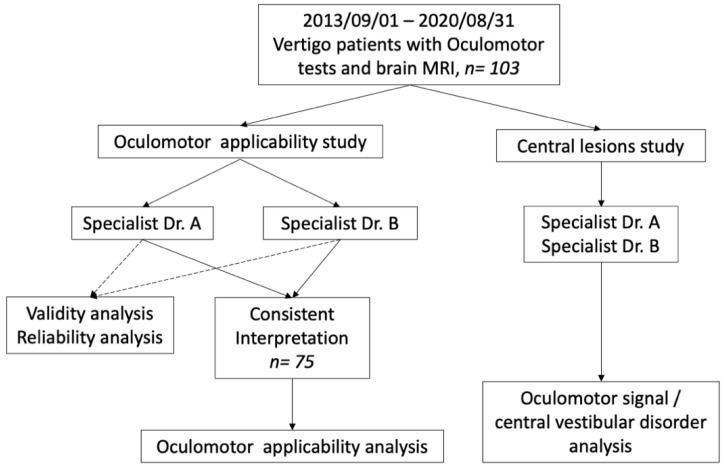
Flow diagram of the overall study design.

**Figure 2 jpm-12-00203-f002:**
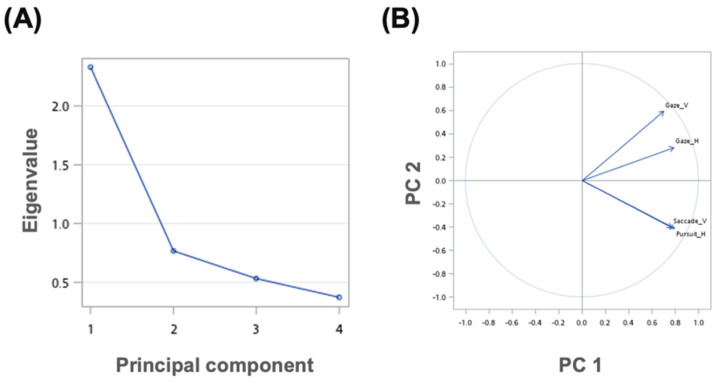
Principal component analysis of oculomotor tests. (**A**) The scree plot shows that the anchor of the graph where the eigenvalues appear to level off lies in PC2. (**B**) The biplot shows the projections of the oculomotor tests on PC1 and PC2 with the four significantly predictive dimensions. PC = principal component.

**Table 1 jpm-12-00203-t001:** Demographic and clinical characteristics of the 103 vertigo patients included in the study.

Variables	All Patients *n* = 103	Central *n* = 24	Nonspecific *n* = 79	*p*-Value
**Median age at diagnosis,**				0.794
years (IQR)	60 (49–69)	61 (48–69)	60 (49–69)	
**Gender**				* 0.014
Female	65 (63.1%)	10 (41.7%)	55 (69.6%)	
Male	38 (36.9%)	14 (58.3%)	24 (30.4%)	
**Body mass index (n = 99)**				0.339
Median (IQR)	24.0 (21.7–25.7)	24.7 (22.5–26.6)	23.8 (21.7–25.7)	
**Asymmetric hearing loss (n = 85)**				0.379
No	71 (83.5%)	11 (91.7%)	60 (82.2%)	
Yes (>30 dB loss)	14 (16.5%)	1 (8.3%)	13 (17.8%)	
**Neurologic symptoms**				* <0.001
No	78 (75.7%)	8 (33.3%)	70 (88.6%)	
Yes	25 (24.3%)	16 (66.7%)	9 (11.4%)	
**Central lesions (MRI)**		N/A	N/A	
No	79 (76.7%)			
Yes	24 (23.3%)			
**Lesion sites (MRI)**		N/A	N/A	
Cortical and subcortical	10 (38.5%)			
Brain stem	10 (38.5%)			
Cerebellar	5 (19.2%)			
Skull base	1 (3.8%)			
**Lesion types (MRI)**		N/A	N/A	
Cerebrovascular accident	13 (54.2%)			
Tumor	9 (37.5%)			
Inflammation	2 (8.3%)			
**Comorbidities**				
Diabetes mellitus				0.556
No	83 (80.6%)	18 (75.0%)	65 (82.3%)	
Yes	20 (19.4%)	6 (25.0%)	14 (17.7%)	
Hypertension				0.323
No	52 (50.5%)	10 (41.7%)	42 (53.2%)	
Yes	51 (49.5%)	14 (58.3%)	37 (46.8%)	
Hyperlipidemia				0.253
No	66 (64.1%)	13 (54.2%)	53 (67.1%)	
Yes	37 (35.9%)	11 (45.8%)	26 (32.9%)	
History of CVA				0.622
No	97 (94.2%)	22 (91.7%)	75 (94.9%)	
Yes	6 (5.8%)	2 (8.3%)	4 (5.1%)	
Cardiovascular disease				0.588
No	98 (95.2%)	24 (100.0%)	74 (93.7%)	
Yes	5 (4.8%)	0 (0.0%)	5 (6.3%)	
Accumulated comorbidities				0.709
0	28 (27.2%)	5 (20.8%)	23 (29.1%)	
1–2	64 (62.1%)	16 (66.7%)	48 (60.8%)	
≥3	11 (10.7%)	3 (12.5%)	8 (10.1%)	

* *p* < 0.05. Abbreviations: CVA, cerebrovascular accident; IQR, interquartile range; N/A, not applicable.

**Table 2 jpm-12-00203-t002:** Validity and reliability of the oculomotor test interpretations of the two specialists.

Raters	Sensitivity	Specificity	PPV	NPV	Test–Retest Reliability Kappa (95% CI)	Inter-Rater Reliability Kappa (95% CI)
Dr. A	54.2%	67.1%	33.3%	82.8%	0.669 (0.521–0.817)	0.480 (0.329–0.630)
Dr. B	66.7%	43.0%	26.2%	81.0%	0.571 (0.413–0.729)	

Abbreviations: NPV, negative predictive value; PPV, positive predictive value.

**Table 3 jpm-12-00203-t003:** Demographic and clinical characteristics of patients whose oculomotor tests were consistent and inconsistent with the MRI findings (*n* = 75).

Variables	Consistent n = 44	Inconsistent n = 31	*p*-Value
**Median age at diagnosis**,			* <0.001
years (IQR)	55 (41.5–63.5)	69 (52–76)	
**Gender**			0.968
Female	30 (68.2%)	21 (67.7%)	
Male	14 (31.8%)	10 (32.3%)	
**Body mass index**			0.378
Median (IQR)	23.2 (20.6–25.5)	24.0 (21.8–25.9)	
**Asymmetric hearing loss**			0.804
No	33 (86.8%)	24 (88.9%)	
Yes (>30 dB loss)	5 (13.2%)	3 (11.1%)	
**Neurologic symptoms**			0.640
No	36 (81.4%)	24 (77.4%)	
Yes	8 (18.6%)	7 (22.6%)	
**Central lesions (MRI)**			0.478
No	34 (77.3%)	26 (83.9%)	
Yes	10 (22.7%)	5 (16.1%)	
**Lesion sites (MRI)**			0.327
Cortical and subcortical	3 (27.3%)	1 (20.0%)	
Brain stem	6 (54.6%)	1 (20.0%)	
Cerebellar	2 (18.1%)	3 (60.0%)	
**Lesion types (MRI)**			0.213
Cerebrovascular accident	7 (70.0%)	2 (40.0%)	
Tumors	3 (30.0%)	1 (20.0%)	
Inflammation	0 (0.0%)	2 (40.0%)	
**Comorbidities**			
Diabetes mellitus			0.468
No	37 (84.1%)	24 (77.4%)	
Yes	7 (15.9%)	7 (22.6%)	
Hypertension			0.204
No	25 (56.8%)	13 (41.9%)	
Yes	19 (43.2%)	18 (58.1%)	
Hyperlipidemia			0.240
No	30 (68.2%)	17 (54.8%)	
Yes	14 (31.8%)	14 (45.2%)	
History of CVA			0.067
No	43 (97.7%)	27 (87.1%)	
Yes	1 (2.3%)	4 (12.9%)	
Cardiovascular disease			0.160
No	43 (97.7%)	28 (90.3%)	
Yes	1 (2.3%)	3 (9.7%)	
Accumulated comorbidities			* 0.017
0	15 (34.1%)	3 (9.7%)	
1–2	27 (61.3%)	23 (74.2%)	
≥3	2 (4.6%)	5 (16.1%)	

* *p* < 0.05. Abbreviations: CVA, cerebrovascular accident; IQR, interquartile range.

**Table 4 jpm-12-00203-t004:** Univariate and multivariate analyses of factors predicting discordance between the interpretations of oculomotor tests and brain MRI (*n* = 75).

Variables	Univariate OR (95% CI)	*p*-Value	Multivariate OR (95% CI)	*p*-Value
**Age**				
<60 years old (n = 34)	1		1	
≥60 years old (n = 41)	4.15 (1.52–11.34)	* 0.006	3.09 (1.04–9.14)	* 0.042
**Gender**				
Female	1		1	
Male	1.02 (0.38–2.73)	0.968	0.91 (0.30–2.83)	0.877
**Body mass index (n = 74)**	1.05 (0.94–1.17)	0.407		
**Asymmetric hearing loss**				
No (n = 57)	1			
Yes (>30 dB loss) (n = 8)	0.83 (0.18–3.79)	0.805		
**Neurologic symptoms**				
No	1		1	
Yes	1.31 (0.42–4.10)	0.640	0.99 (0.28–3.56)	0.991
**Central lesions (MRI)**				
No	1			
Yes	0.65 (0.20–2.15)	0.484		
**Comorbidities**				
Diabetes mellitus				
No	1			
Yes	1.54 (0.48–4.95)	0.467		
Hypertension				
No	1			
Yes	1.82 (0.72–4.72)	0.206		
Hyperlipidemia				
No	1			
Yes	1.77 (0.68–4.56)	0.241		
History of CVA				
No	1			
Yes	6.37 (0.68–60.05)	0.106		
Cardiovascular disease				
No	1			
Yes	4.61 (0.46–46.54)	0.195		
Accumulated Comorbidities				
0	1		1	
1–2	4.26 (1.10–16.57)	* 0.037	2.89 (0.65–12.92)	0.164
≥3	12.50 (1.60–97.64)	* 0.016	8.31 (0.96–71.71)	0.054

* *p* < 0.05. Abbreviations: CVA, cerebrovascular accident; OR, odds ratio.

**Table 5 jpm-12-00203-t005:** Univariate analysis of oculomotor signals predicting abnormal brain MRI. The “factor score” was calculated via principal component analysis of significantly predictive tests (*n* = 103).

Oculomotor Signals	Univariate OR (95% CI)	*p*-Value	Factor Score
**Saccade**			
Horizontal			
No	1		
Yes	1.85 (0.71–4.83)	0.206	
Vertical			
No	1		
Yes	3.58 (1.29–9.98)	* 0.015	0.4981
**Pursuit**			
Horizontal			
No	1		
Yes	3.39 (1.26–9.09)	* 0.016	0.5120
Vertical			
No	1		
Yes	2.25 (0.84–6.03)	0.106	
**Gaze**			
Horizontal			
No	1		
Yes	3.06 (1.19–7.85)	* 0.020	0.5228
Vertical			
No	1		
Yes	4.72 (1.72–12.99)	* 0.003	0.4653
**OPK**			
No	1		
Yes	1.22 (0.44–3.36)	0.707	
**OKAN**			
No	1		
Yes	2.32 (0.83–6.48)	0.108	

* *p* < 0.05. “no”: no signal was labeled by both raters; “yes”: signal was labeled by either rater. Abbreviations: OKAN, optokinetic afternystagmus; OPK, optokinetic nystagmus; OR, odds ratio.

**Table 6 jpm-12-00203-t006:** Univariate and multivariate analyses of factors predicting abnormal brain MRI (*n* = 103).

Variables	Univariate OR (95% CI)	*p*-Value	Multivariate OR (95% CI)	*p*-Value
**Age**				
<60 years-old (n = 48)	1		1	
≥60 years-old (n = 55)	1.30 (0.52–3.27)	0.581	0.73 (0.20–2.68)	0.632
**Gender**				
Female	1		1	
Male	3.21 (1.25–8.23)	* 0.015	2.85 (0.82–9.90)	0.100
**Body mass index (n = 99)**				
Median (IQR)	1.03 (0.93–1.15)	0.528		
**Asymmetric hearing loss (n = 85)**				
No	1			
Yes (>30 dB loss)	0.42 (0.05–3.54)	0.425		
**Neurologic symptoms**				
No	1		1	
Yes	15.56 (5.20–46.56)	* <0.001	13.45 (4.00–45.12)	* <0.001
**Comorbidities**				
Diabetes mellitus				
No	1			
Yes	1.55 (0.52–4.60)	0.432		
Hypertension				
No	1			
Yes	1.59 (0.63-4.00)	0.326		
Hyperlipidemia				
No	1			
Yes	1.73 (0.68-4.37)	0.251		
History of CVA				
No	1			
Yes	1.71 (0.29-9.94)	0.553		
**Accumulated comorbidities**				
0	1		1	
1–2	1.53 (0.50–4.70)	0.455	0.56 (0.11–2.79)	0.481
≥3	1.73 (0.33–8.91)	0.515	0.47 (0.05–4.44)	0.506
**Oculomotor index**				
<50%	1		1	
>50%	4.65 (1.66–12.99)	* 0.003	4.59 (1.28–16.44)	* 0.019

* *p* < 0.05. Abbreviations: CVA, cerebrovascular accident; IQR, interquartile range; OR, odds ratio.

**Table 7 jpm-12-00203-t007:** Validity analysis of oculomotor tests according to age and comorbidities in both raters.

Raters	Sensitivity	Specificity	PPV
**Dr. A**			
<60 years old	60.0%	84.2%	50.0%
≥60 years old	50.0%	51.2%	25.9%
**Dr. B**			
<60 years old	60.0%	57.9%	27.3%
≥60 years old	71.4%	29.3%	25.6%
**Dr. A**			
Comorbidities: 0	40.0%	100.0%	100.0%
Comorbidities: 1, 2	62.5%	56.3%	32.3%
Comorbidities: ≥3	33.3%	37.5%	16.7%
**Dr. B**			
Comorbidities: 0	0.0%	65.2%	0.0%
Comorbidities: 1, 2	87.5%	35.4%	31.1%
Comorbidities: ≥3	66.7%	25.0%	25.0%

Abbreviations: PPV, positive predictive value.

## Data Availability

The data presented in this study are available on request from the corresponding author. The data are not publicly available because all the original MRI and oculomotor data are in company with an identifiable chart number.

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
