# Peer review of "Applicability of Oculomotor Tests for Predicting Central Vestibular Disorder Using Principal Component Analysis"

_jpm, 2022, doi:10.3390/jpm12020203_

Round 1
Reviewer 1 Report
Useful study
Author Response
Thanks for your hard work.
Reviewer 2 Report
The present paper discusses data obtained in 103 patients with vertigo or dizziness symptoms. The aim of the research was to evaluate the utility of oculomotor tests for diagnosing central vestibular disorder. Results obtained in oculomotor tests were correlated with brain MRI. The experiments were carefully planned and results showed that particularly for patients older than 60 this test is not highly predictive. In Youngers subjects neurological symptoms and abnormal oculomotor index suggest central lesions. Statistical analysis appears appropriate. Supplementary table might be included.
Author Response
Thank you for your hard work and detailed review. We included the supplementary table into table 7 as your suggestion.
Reviewer 3 Report
Summary: The authors investigated the utility of oculomotor tests for diagnosing central vestibular disorder and the correlations between oculomotor tests and brain MRI findings. The study provided various critical findings bringing out the factors associated with central vestibular disorder, oculomotor analysis techniques and correlation with brain MRI features which makes it interesting for the research field. The major concerns of the study are lack of details related to methods and results as listed below.
Comments:
1. Introduction:
Page 2, Line 59: The authors mentioned about the cons of using imaging diagnosis for central vestibular disorder stating that "routine MRI" is unhelpful and unnecessary for determining the specific cause of dizziness or vertigo and hence it has to be accompanied with vestibular function tests. Is it a statement to indicate that specific contrasts of MRI are required? More details justifying how MRI has been used in vertigo earlier along with pros and cons would add more justification to the aim of the study.
2. Methods:
Oculomotor tests: Details regarding the number of measurements involved in each of the tests needs to be elaborated with the units of measurements and the plausible range of values associated with each of them as the description looks vague in the current form.
Brain MRI:
- The manuscript lacks MRI acquisition and protocol details. Details such as scanner parameters, pulse sequence and imaging parameters are must for anyone who wants to reproduce the study.
- MRI has various contrasts such as T1w, T2w, diffusion, functional etc. It is unclear from the methods section as to what kind of contrasts images are used for the diagnosis
- what is the imaging features obtained from MRI which resulted in identifying the diseased subjects, is it visual or using any scientific parameter estimation, if so, what is the technique or models used?
Data Analysis:
- what is the reason behind using both univariate and multivariate in parallel for all statistical tests, the authors need to justify the reason behind selection of the tests?
- line 148: how many oculomotor signals are measured for a subject, what is the dimensions of the signals
- line 151 says high correlation between these signals, which signal are they, is it the signals between subjects or is the correlation between different signals of a subject, if so what are those signals and how many signals or measurements taken. These details would give the readers clear idea about the dimensions of the data on which PCA is applied
- PCA implementation and oculomotor index calculation is not clear, for eg, how many principal components were estimated, the original size of the signals, the size of the mixing coefficients and the reduced data are unknown
Results:
Table 1: how is the central (n=24) and non-specific(n=79) group classified? Line 170 indicating "based on results of brain MRI" is not clear. Please elaborate. Also include images for both groups showing lesions in abnormal and normal tissues in corresponding normal group.
Line 190: "In total, 75 such tests were analyzed" - is the 75 correspond to subjects or various test measurements?
Lines 209-220:
Add details regarding the nature of the signals and how correlated are they to justify adopting PCA
Figure 2B) why OPK and OKAN are missing, are they not PCs?
